# Cobalt Ferrite Synthesized Using a Biogenic Sol–Gel Method for Biomedical Applications

**DOI:** 10.3390/molecules28237737

**Published:** 2023-11-23

**Authors:** Patrícia Gomes, Bárbara Costa, João P. F. Carvalho, Paula I. P. Soares, Tânia Vieira, Célia Henriques, Manuel Almeida Valente, Sílvia Soreto Teixeira

**Affiliations:** 1i3N and Department of Physics, University of Aveiro, 3810-193 Aveiro, Portugal; patriciafjgomes@ua.pt (P.G.); barbaracostaa@ua.pt (B.C.); jpfc@ua.pt (J.P.F.C.); mav@ua.pt (M.A.V.); 2CENIMAT, Department of Materials Science, School of Science and Technology, NOVA University Lisbon, 2829-516 Caparica, Portugal; pi.soares@fct.unl.pt; 3CENIMAT/i3N, Department of Physics, School of Science and Technology, NOVA University Lisbon, 2829-516 Caparica, Portugal; ts.vieira@fct.unl.pt (T.V.); crh@fct.unl.pt (C.H.)

**Keywords:** magnetic nanoparticles, cobalt ferrite, sol–gel, cancer, magnetic hyperthermia

## Abstract

Cancer is one of the leading causes of death worldwide. Conventional treatments such as surgery, chemotherapy, and radiotherapy have limitations and severe side effects. Magnetic hyperthermia (MH) is an alternative method that can be used alone or in conjunction with chemotherapy or radiotherapy to treat cancer. Cobalt ferrite particles were synthesized using an innovative biogenic sol–gel method with powder of coconut water (PCW). The obtained powders were subjected to heat treatments between 500 °C and 1100 °C. Subsequently, they were characterized by thermal, structural, magnetic, and cytotoxic analyses to assess their suitability for MH applications. Through X-ray diffraction and Raman spectroscopy, it was possible to confirm the presence of the pure phase of CoFe_2_O_4_ in the sample treated at 1100 °C, exhibiting a saturation magnetization of 84 emu/g at 300 K and an average grain size of 542 nm. Furthermore, the sample treated at 1100 °C showed a specific absorption rate (SAR) of 3.91 W/g, and at concentrations equal to or below 5 mg/mL, is non-cytotoxic, being the most suitable for biomedical applications.

## 1. Introduction

Despite medical advances, cancer remains one of the most prevalent diseases in the world, with a high mortality rate. According to the Global Cancer Observatory (GCO) [1], in 2020, more than 19 million new cancer cases and almost 10 million deaths were registered. The field of nanomedicine has emerged with the intent of overcoming the existing issues of conventional treatments [2,3]. In recent years, this area has shown significant advances in the treatment and diagnosis of cancer. Magnetic nanoparticles (MN) are nanostructures with magnetic properties with at least one dimension at the nanoscale [4,5] that have been exploited in magnetic hyperthermia (MH) therapy, a non-invasive cancer treatment. To be applied for biomedical purposes, they must be biocompatible, non-toxic, and be sufficiently small (between 1 and 100 nm) [6] to preserve colloidal stability [4,5,7].

Magnetic hyperthermia is an approach that involves the administration of magnetic nanoparticles directly to the tumor site followed by the application of an alternating magnetic field, increasing the temperature of the nanoparticles located near the cancer cells. This approach allows these cells to be heated between 41 and 46 °C [8] while preserving surrounding normal tissue, potentially improving the efficacy and safety of hyperthermia. The particles exposed to an alternating magnetic field transfer heat through Néel and Brownian relaxations [5,9], defined by the specific absorption rate (SAR). 

Ferrites are composed of iron oxides, and the ones with a spinel structure stand out since they are already being used in the separation and purification of cells, as well as in drug delivery and as contrast agents for magnetic resonance imaging (MRI) and magnetic hyperthermia. Ferrites have a general molecular formula given by AB_2_O_4_, with A and B corresponding to transition metals [10]. Due to the continuous need to synthesize new efficient magnetic nanoparticles, to be used for magnetic hyperthermia, the recent literature describes a range of magnetic nanoparticles, from Fe_3_O_4_ with different shapes [11,12,13,14,15] to LiFe_5_O_8_ [5] and CoFe_2_O_4_@Ni_0.5_Zn_2.6_ Fe_2_O_4_ [16]. Recently, an article described a significant increase in SAR values with Zn-substituted magnetite superparamagnetic nanoparticles [17]. Therefore, the necessity for the scientific community to develop new ferrites with the proper thermal efficiencies to be used for magnetic hyperthermia is notable.

CoFe_2_O_4_ has an inverted spinel structure where the Co^2+^ ions occupy only octahedral sites and the Fe^3+^ ions are equally distributed between the tetrahedral and octahedral locations. This distribution of ions is crucial because it influences their magnetic interaction, which affects their magnetic properties. In addition, it can also affect electrical properties such as electrical conductivity and dielectric characteristics [18].

Cobalt ferrite is a material that has been extensively studied due to its high electromagnetic performance, excellent chemical stability, mechanical hardness, high coercivity (4.3 kOe at 300 K), high anisotropy, high Curie temperature (520 °C), and high saturation magnetization (80 emu/g). The properties of cobalt ferrite make it one of the most promising materials for medical applications, including radiofrequency hyperthermia and MRI [18,19].

The objectives of this study encompassed the synthesis, physical characterization, and assessment of the potential applicability of cobalt ferrite nanoparticles in magnetic hyperthermia to be effectively used for cancer treatment. Cobalt particles were synthesized employing a sol–gel method utilizing a protein-mediated process using powder of coconut water (PCW) as a precursor. According to the necessity of employing green synthesis methods in science, PCW was chosen as the precursor for the sol–gel reaction. Consisting of a rich composition of sugar, amino acids, and minerals, this compound acts as a surfactant for the reaction [5,20]. This approach has been employed previously for the synthesis of nanoparticles such as BaFe_12_O_19_ [21], SrFe_12_O_19_ [22], KfeO_2_ [23], and SrAl_2_O_4_ [24].

Through this study, we intended to assess the impact of various parameters related to thermal treatment on distinct aspects encompassing the formation of the cobalt ferrite phase, and the granularity, magnetic attributes, cytotoxicity, and SAR of the particles. This work intended to synthesize cobalt ferrite using an innovative sol–gel method, using PCW as the surfactant. These particles are intended to be used for cancer treatment through magnetic hyperthermia.

## 2. Results

### 2.1. Thermal Analysis

DTA and TG were performed to determine the THT temperatures to be applied to the samples.

Thermal analysis was performed on the powders obtained after the synthesis reaction and solvent evaporation. It was possible to identify two relevant exothermic transformations, one at approximately 623.15 K and the other at 1123.15 K (Figure 1). The loss between 423.15 K and 623.15 K can be attributed to the loss of absorbed water and the dehydration of hydroxyl groups (OH^−^), and the loss between 636.15 K and 1098.15 K can be ascribed to the decomposition of chlorides and organic chains derived from ACP.

The DTA measurements identified two relevant exothermic transformations, one at 623.15 K and the other at 1123.15 K. These transformations suggest the formation of new crystalline phases, which may include the $Co{Fe}_{2}O_{4}$ phase.

Considering the DTA/TG results, thermal heat treatments (HTs) were applied at 773.15 K, 973.15 K, 1073.15 K, and 1173.15 K. Thus, the samples were HT in an oven (Termolab, Aveiro, Portugal) at the desired temperature for 4 h. To clarify the temperature of the THTs, the samples are referred to as 500 °C, 700 °C, 900 °C, 1000 °C, and 1100 °C.

### 2.2. Structural Characterization

#### 2.2.1. XRD Diffractograms

Figure 2 shows the X-ray diffraction (XRD) results of the samples subjected to different THTs; Table 1 presents the weight percentages (wt%) of the phases present in each sample, obtained through the Rietveld refinement technique; Figure 3 shows the goodness-of-fit (GF) value. GF values close to 1 indicate good agreement between the experimental data and the expected theoretical patterns [25].

The diffractogram obtained at 1100 °C agrees with the cobalt ferrite pattern reported in the literature [26,27] since the peaks show similar correspondence regarding their position and relative intensity.

After the thermal tests were performed, cobalt ferrite was observed, except for the sample HT at 500 °C. This indicates that after the HT temperature of 500 °C there was a change in the composition or structure of the sample, leading to a shift in the predominant crystalline phase.

In the sample HT at 500 °C, 77.3% of γ-${\mathrm{Fe}}_{2}O_{3}$ and 22.7% of ${\mathrm{Co}}_{3}O_{4}$ were obtained based on mass percentage (% m/m). In the sample HT at 700 °C, a change from γ-${\mathrm{Fe}}_{2}O_{3}$ to α-${\mathrm{Fe}}_{2}O_{3}$ was verified, which agrees with the literature [28]. An increase in the mass percentage of ${\mathrm{Co}}_{3}O_{4}$ was also observed, which can be attributed to the crystallization of the ${\mathrm{Co}}_{3}O_{4}$ that was amorphous in the sample HT at 500 °C.

For the sample HT at 900 °C, there was a significant increase in the mass percentage of $\mathrm{Co}{\mathrm{Fe}}_{2}O_{4}$, with up to 94.7% of the sample attributed to the reaction between α-${\mathrm{Fe}}_{2}O_{3}$ and the amorphous phase of ${\mathrm{Co}}_{3}O_{4}$. For the sample HT at 1000 °C, increases in the percentages of cobalt ferrite and α-${\mathrm{Fe}}_{2}O_{3}$ were still detected. By increasing the temperature to 1100 °C, the α-${\mathrm{Fe}}_{2}O_{3}$ phase was no longer detected, resulting in a sample containing only $\mathrm{Co}{\mathrm{Fe}}_{2}O_{4}$ with a crystallite size of 131.7 nm. This transition suggests a complete reaction between the amorphous crystalline phases of ${\mathrm{Co}}_{3}O_{4}$ and α-${\mathrm{Fe}}_{2}O_{3}$.

The obtained GF value of the sample HT at 1100 °C is close to 1, indicating good agreement with the experimental data.

#### 2.2.2. Raman Spectroscopy

Cobalt ferrite has a cubic spinel crystalline structure and belongs to the Fd-3m group. According to the literature, it is expected to present 39 distinct vibration bands; however, in the Raman spectroscopy results, only five of these bands were observed, between 648 and 680 cm^−1^, 278 and 293 cm^−1^, 539 and 565 cm^−1^, 449 and 471 cm^−1^, and 163 and 177 cm^−1^ [29]. The results obtained from the XRD analysis (Figure 3) corroborated the Raman spectroscopy results (Figure 4), except for the samples treated at 900 °C and 1000 °C, as it was possible to identify the band associated with ${Co}_{3}O_{4}$. However, this band was not identified in the XRD spectra since ${Co}_{3}O_{4}$ does not have sufficient crystallinity to be detected. Furthermore, it was possible to identify the bands associated with ${Co}_{3}O_{4}$ and γ-${Fe}_{2}O_{3}$ in the sample HT at 500 °C; the spectra of the samples heat treated at 700 °C, 900 °C, and 1100 °C supported the identification of phases of ${Co}_{3}O_{4}$, $Co{Fe}_{2}O_{4}$, and α-${Fe}_{2}O_{3}$; and in the sample heat treated at 1100 °C, only the band associated with cobalt ferrite was identified (Table 2).

### 2.3. Morphological Characterization

Scanning electron microscopy (SEM), when associated with an energy-dispersive X-ray spectrometer (EDS), allows the elemental composition of each sample to be identified. Figure 5 shows the SEM results of the samples HT at 700 °C, 900 °C, and 1100 °C. The sample HT at 700 °C shows a high level of agglomeration, and, in addition, has three crystalline phases present (Figure 2), making the identification of $Co{Fe}_{2}O_{4}$ grains difficult; therefore, the grain size of this sample was not determined. With an increase in the HT temperature from 900 °C to 1100 °C, the average grain size increased from 458 nm to 542 nm, respectively. For the sample HT at 900 °C and 1100 °C, a high degree of heterogeneity can be seen in the particle-size distributions.

Given the particle sizes obtained (>100 nm), it can be concluded that rather than magnetic particles, microparticles were obtained. These results show a direct relationship between an increase in the THT temperature and grain growth. The EDS mapping revealed that elements were not uniformly distributed throughout the samples, since it was possible to identify specific regions where the presence of Co or Fe was more predominant, indicating heterogeneity in the distribution of these elements. As described in the literature [35], the image obtained for cobalt ferrite at 1100 °C revealed the presence of agglomerated particles with a morphology that resembles a sponge. This arrangement of clustered particles results in a porous morphology.

### 2.4. Magnetic Characterization

Figure 6 shows the magnetization curves as a function of the magnetic field (M-H curves) for the TT samples HT at 700 °C, 900 °C, and 1100 °C and measured at 5 K and 300 K.

At 5 K, the saturation magnetization of the samples HT at 700 °C, 900 °C, and 1100 °C were 55 emu/g, 88.5 emu/g, and 86.8 emu/g, respectively. A decrease in magnetization was observed between the sample HT at 900 °C and the sample HT at 1100 °C, which is unexpected as the percentage of $\mathrm{Co}{\mathrm{Fe}}_{2}O_{4}$ increases. However, it is essential to consider the error associated with the measuring equipment, indicating that this variation is now significant.

The magnetization of the sample HT at 300 K was 86.5 emu/g, which agrees with the literature [27,34]. Furthermore, the magnetization curve became saturated for magnetic fields greater than 2.1 T. The sample also showed a coercive field of Hc = 0.17 T and a remanent magnetization of Mr = 30 emu/g.

At room temperature, the sample HT at 900 °C showed a saturation magnetization of 79 emu/g, which was lower than the TT of the sample HT at 1100 °C. According to the XRD analysis, this sample contained only $\mathrm{Co}{\mathrm{Fe}}_{2}O_{4}$ and ${\mathrm{Fe}}_{2}O_{3}$. Based on the literature, ${\mathrm{Fe}}_{2}O_{3}$ is expected to have a magnetization of approximately 74 emu/g [36], which explains the observed decrease in magnetization compared to the sample HT at 1100 °C. This sample also showed an Hc value of 0.21 T and an Mr value of 21 emu/g.

The sample HT at 700 °C had a maximum magnetization, at 300 K, of 50 emu/g, which is significantly lower than that observed for the samples HT at higher temperatures. This result confirms the predominant presence of ${\mathrm{Co}}_{3}O_{4}$ in this sample, which has a magnetization of approximately 59 emu/g according to the literature [37]. This sample also had a Hc value of 0.23 T and a Mr value of 18 emu/g. It can be seen, therefore, that for all samples, the coercive field (Hc) and remanent magnetization (Mr) decreased from the 5 K to 300 K analyses. This increase in the temperature induces an increase in atom/electron agitation, increasing disorder and making the alignment of magnetic dipoles with the external magnetic field more difficult, explaining the Mr reduction for the measurement at 300 K. On the other hand, at a lower temperature (5 K), the magnetic dipoles are more fixed in their orientation, making reorientation according to the external field more difficult and resulting in lower Hc values at this temperature.

### 2.5. Specific Absorption Rate

SARs were evaluated to verify the suitability of the samples for being employed in MH treatments. Table 3 represents the obtained results.

When evaluating the SAR values, we found that the samples HT at 700 °C, 900 °C, and 1100 °C exhibited values of 1.10 W/g, 2.49 W/g, and 3.91 W/g, respectively, which indicates that the SAR increases as the cobalt ferrite phase increases. Although higher sintering temperatures cause a slight increase in warming, the SAR remains low. Despite the high saturation magnetization, the SAR is still low. This can be attributed to the particles which do not achieve their superparamagnetic state since they have a higher average size. Further studies with a reduced particle size should be conducted. Considering the SAR at different THTs, no sample would be appropriate for magnetic hyperthermia.

### 2.6. Cytotoxicity

The results obtained from the cell viability assays are illustrated in Figure 7, and the analysis of these results was carried out under the guidelines established by ISO 10993-5:2009 [38]. As stipulated, samples where the relative cell viability was higher than 80% do not demonstrate cytotoxic effects and, therefore, are considered suitable for biomedical applications [38].

It was found that all samples with a concentration of 20 mg/mL and 10 mg/mL were cytotoxic, regardless of the thermal treatment to which they were submitted. Samples HT at 700 °C and 900 °C were non-cytotoxic only at the lowest concentration of 1.125 mg/mL. Finally, the samples HT at 1100 °C were non-cytotoxic at concentrations equal to or less than 5 mg/mL. These results suggest that the samples treated at 1100 °C can be safely used for biomedical applications without cytotoxic effects.

## 3. Materials and Methods

### 3.1. Cobalt Ferrite Powder Preparation

The synthesis of cobalt ferrite powder was carried out through a protein-based sol–gel pathway. For the synthesis of the cobalt ferrite powder, a stoichiometric ratio was used: 1 mol nitrate ${\left(Fe({NO}_{3}\right)}_{3}.9H_{2}O)$ (Sigma-Aldrich, St. Louis, MO, USA; ≥98%): 1 mol cobalt nitrate ${\left(Co({NO}_{3}\right)}_{2}.6H_{2}O)$ (Sigma-Aldrich; 98%).

To obtain 30 g of cobalt ferrite, 47.90 g of powder of coconut water (*Cocos nucifera* L.) was dissolved in 500 mL of deionized water. Thus, a solution of PCW with a concentration of 0.58 mol·${dm}^{-3}$, the critical micellar concentration, was obtained, which facilitated the enhanced dispersion and uniformity of the solution [39]. The raw materials were then blended using a magnetic stirrer (C-MAG HS4 digital, Ika, Staufen, Germany) using the following parameters: (i) temperature (T) = 80 °C, time (∆*t*) = 2 h; and (ii) T = 100 °C, ∆*t* = 1 h, to produce a gel with high viscosity.

The gel was heat-treated (HT) at 350 °C in an oven (Termolab, Aveiro, Portugal) for one hour to eliminate the solvents and obtain a dry powder, using a heating rate of 5 °C/min for 1 h. The resultant powder was molded into disk-like shapes, which were HT from 500 to 1100 °C.

### 3.2. Structural and Morphological Characterization

Thermal differential analysis and thermogravimetric analysis (DTA/TGA) were conducted using STA7300 equipment (Hitachi, Fukuoka, Japan) under a nitrogen atmosphere with a 200 mL flux. The analyses were conducted with a heating rate of 5 K/min from room temperature to 1250 K, under a nitrogen atmosphere. The samples were placed in an alumina (Al_2_O_3_) crucible. The sample measurements were compared with alumina, enabling the data to be obtained.

The powder X-ray diffraction (XRD) was performed using an AERIS diffractometer (Malvern Panalytical, Malvern, UK). CuK α radiation was employed with an approximate wavelength of 1.54060 Å, operating at a voltage of 45 kV and a current of 40 mA. To acquire the intensity data, a step counting method (0.02° s^−1^) was used in the 2θ angle range of 10 to 60°. X’Pert HighScore Panalytical software was used in conjunction with the Joint Committee for Powder Diffraction Standards–International Center for Diffraction Data (JCPDS) database to identify the crystalline phases.

Raman spectroscopy utilizing a Jobin Yvon spectrometer (Horiba Scientific, Kyoto, Japan) was performed at room temperature using a backscattering geometry setup. A microscope objective (50×) was employed to focus the excitation light from the laser (λ = 532 nm) onto the sample (spot diameter < 0.8 μm).

Scanning Electron Microscopy (SEM) was performed using a Vegan 3 TESCAN microscope (TESCAN ORSAY HOLDING, Brno, Czech Republic), and the average grain size was obtained using ImageJ 1.51 software.

### 3.3. Magnetic Characterization

The magnetic measurements were made using a Cryofree vibrating sample magnetometer (VSM) (Cryogenic Ltd., London, UK). Magnetic curves were obtained at 5 and 300 K using a magnetic field (H) of up to 5 T.

The specific absorption rate (SAR) was measured using a DM100 Series (nB nanoscale Biomagnetics, Zaragoza, Spain) with a 10 mg/mL sample concentration. For this, the samples were subjected to an alternating external magnetic field with a magnitude of 24 kA/m and a frequency of 418.5 kHz for 10 min. Before each measurement, the samples were immersed in 1 mL of ultrapure water and subjected to ultrasonication.

### 3.4. Biological Analysis

Cytotoxicity assays were conducted using the extract method following the ISO 10993-5:2009 International Standard. The Saos-2 cell line from ATCC HTB-85 was obtained from an osteosarcoma patient following the expansion and maintenance protocols provided by the supplier. The extracts, with an initial concentration of 20 mg/mL for each sample, were prepared using a 48 h incubation in McCoy-5A culture medium (BioWest, Riverside, MO, USA) supplemented with 10% fetal bovine serum (BioWest) and 1% penicillin/streptomycin (Gibco, Waltham, MA, USA) at 310.15 K. Saos-2 cells were seeded onto 96-well plates at a density of 30,000 cells/cm^2^ in a complete McCoy-5A culture medium. The cells were maintained at 310.15 K in a 5% CO_2_ atmosphere for 24 h. After that time, the culture medium was replaced by the extract, which was diluted in the culture medium to obtain five concentrations, 20, 10, 5, 2.5, and 1.125 mg/mL, with each concentration tested four times. A negative control was established containing cells maintained in a complete culture medium, and a positive control was established with cells treated with 10% DMSO, a recognized cytotoxic compound. After a 48 h incubation period, the culture medium was carefully removed from each well and replaced with a resazurin (Alfa Aesar, Ward Hill, MA, USA) solution composed of 50% complete culture medium and 50% resazurin solution at a concentration of 0.04 mg/mL in phosphate-buffered saline. Next, the plates underwent an additional 3 h incubation at 310.15 K in a controlled environment with a 5% CO_2_ atmosphere. The absorbance of the samples was measured at two wavelengths using a Biotek Elx 800UV Microplate Reader (Winooski, VT, USA), at 570 nm and 600 nm, and the corrected absorbance was proportional to cellular viability. Viability was expressed as the percentage of viable cells relative to the negative control.

## 4. Conclusions

This project synthesized pure cobalt ferrite particles using a sol–gel method with powder of coconut water solution as a precursor, as a new green synthesis method. To obtain $Co{Fe}_{2}O_{4}$, different heat treatments were applied to the samples. Based on XRD and Raman spectroscopy, it was confirmed that the sample HT at 1100 °C had a pure composition of $Co{Fe}_{2}O_{4}$. Regarding all samples, the obtained particles had average particle sizes of 458 nm (HT = 900 °C) and 542 nm (HT = 1100 °C). Therefore, none of the synthesized samples are suitable for application in a biological environment. In terms of the SAR values obtained for the samples, these were lower than expected. Regarding cytotoxicity, the samples showed non-cytotoxic behavior at a concentration of 1.125 mg/mL, except for the sample HT at 1100 °C, which was considered non-cytotoxic at concentrations lower than 10 mg/mL. In conclusion, among the synthesized samples, the sample HT at 1100 °C presented high saturation magnetization (86.8 emu/g), high biocompatibility, and a pure composition of $Co{Fe}_{2}O_{4}$. The low SAR value of $Co{Fe}_{2}O_{4}$ particles can be attributed to their magnetic (not superparamagnetic) behavior, as a consequence of their high grain size. In future work, it is necessary to reduce the size of the particles using, for example, planetary ball-milling or via synthesizing the material using another method, for instance, co-precipitation.

## Figures and Tables

**Figure 1 molecules-28-07737-f001:**
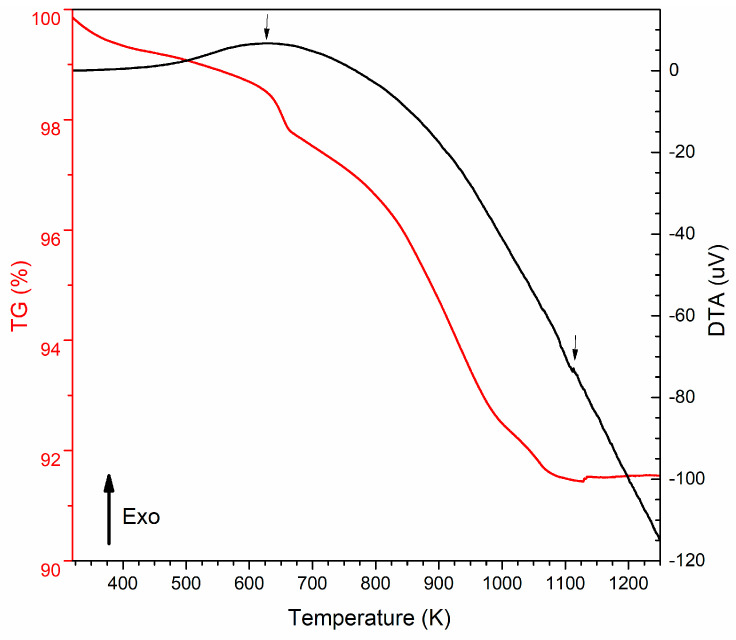
DTA (black curve) and TG (red curve) vs. temperature of the powders obtained after the synthesis reaction.

**Figure 2 molecules-28-07737-f002:**
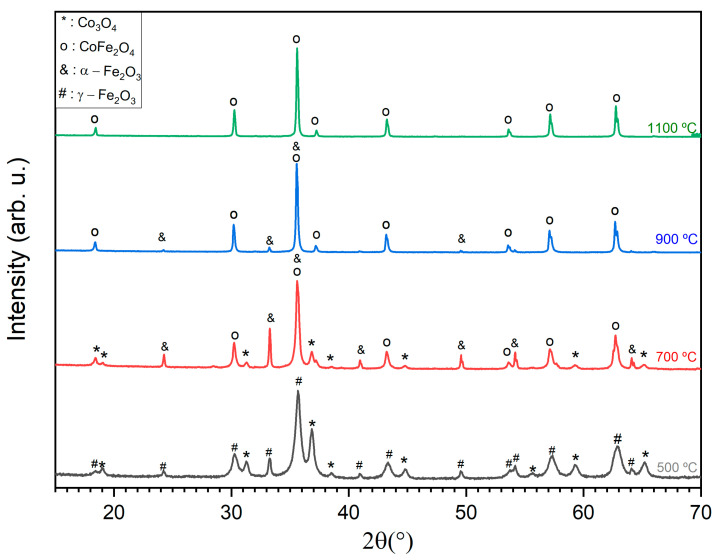
X-ray diffractograms of samples heat treated at different temperatures.

**Figure 3 molecules-28-07737-f003:**
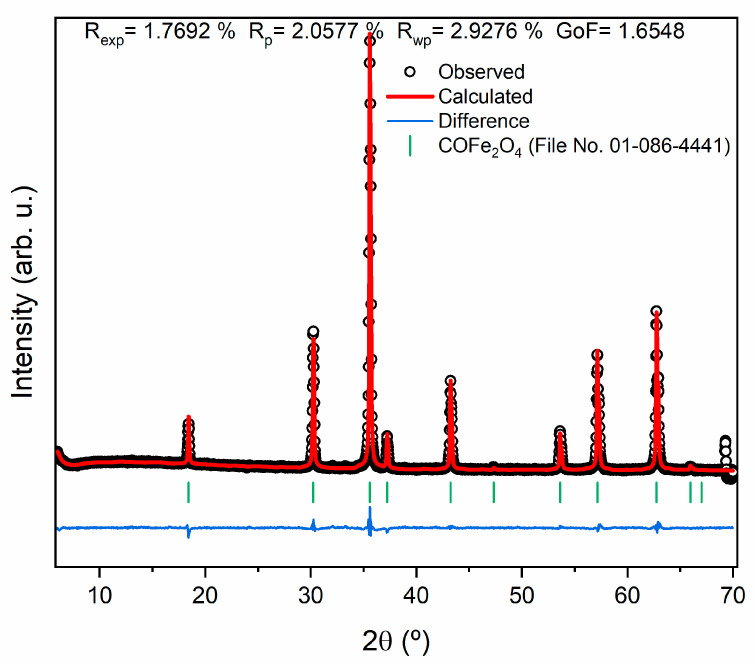
Rietveld refinement of the sample of CoFe_2_O_4_ HT at 1100 °C.

**Figure 4 molecules-28-07737-f004:**
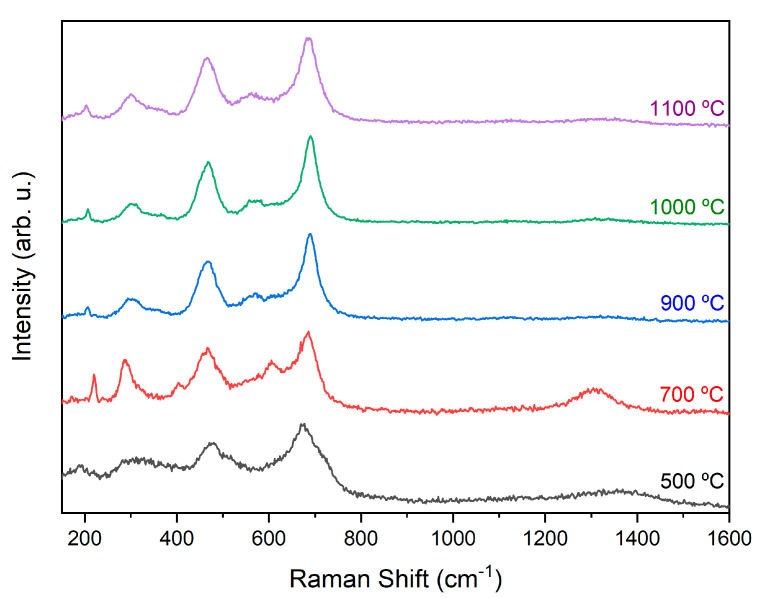
Raman spectroscopy of samples HT at different temperatures.

**Figure 5 molecules-28-07737-f005:**
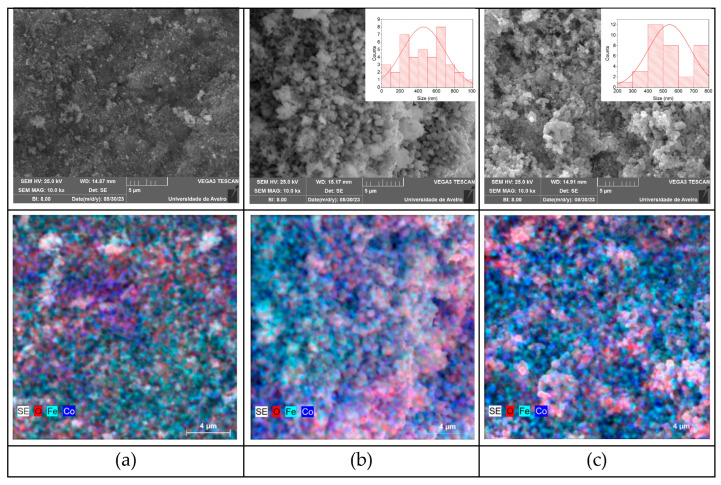
SEM-EDS images of samples HT at (**a**) 700 °C, (**b**) 900 °C, and (**c**) 1100 °C; samples HT at 900 °C and 1100 °C show the grain-size distribution (inset), with an average grain size of 458 nm (900 °C) and 542 nm (1100 °C).

**Figure 6 molecules-28-07737-f006:**
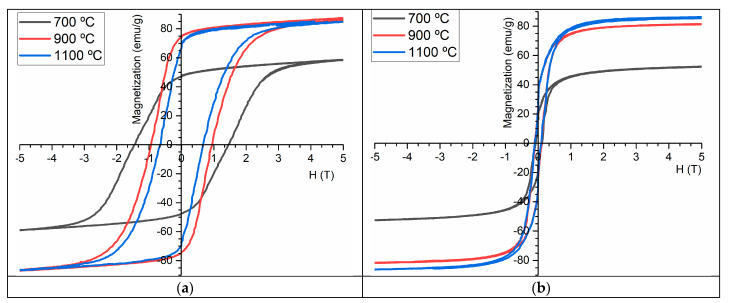
Magnetization vs. magnetic field at (**a**) 5 K and (**b**) 300 K for samples HT at 700 °C, 900 °C, and 1100 °C.

**Figure 7 molecules-28-07737-f007:**
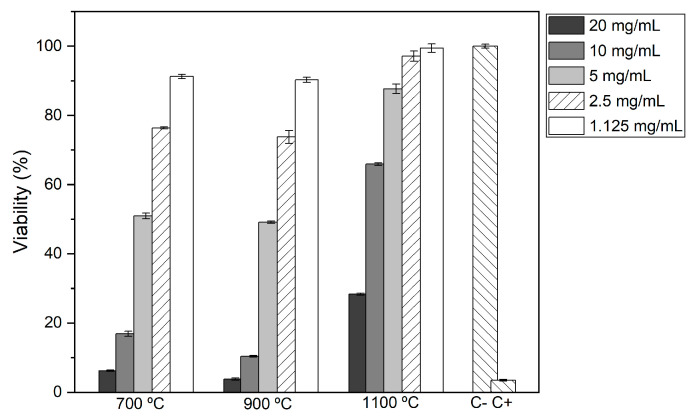
Relative cell viability for different concentrations of each sample.

**Table 1 molecules-28-07737-t001:** Weight percentages (wt%) of the crystalline phases present in each of the samples, obtained through Rietveld refinement.

HT Temperature (°C)	Crystalline Phase and Composition (wt%)	R Expected (R_exp_)	Weighted R Profile (R_wp_)	Goodness-of-Fit (GF)
500	${Co}_{3}O_{4}$ (22.7) γ-${Fe}_{2}O_{3}$(77.3)	1.82	2.84	1.56
700	${Co}_{3}O_{4}$ (63.8) α-${Fe}_{2}O_{3}$ (23.6) $Co{Fe}_{2}O_{4}$ (7.6)	1.69	6.13	3.63
900	$Co{Fe}_{2}O_{4}$ (94.7) α-${Fe}_{2}O_{3}$ (5.3)	1.68	3.82	2.27
1000	$Co{Fe}_{2}O_{4}$ (97.6) α-${Fe}_{2}O_{3}$ (2.4)	1.66	3.64	2.18
1100	$Co{Fe}_{2}O_{4}$ (100)	1.77	2.93	1.65

**Table 2 molecules-28-07737-t002:** Vibrational bands associated with γ-${Fe}_{2}O_{3}$ [28,30], α-${Fe}_{2}O_{3}$ [28,31], ${Co}_{3}O_{4}$ [32,33], and $Co{Fe}_{2}O_{4}$ [27,34].

	500 °C	700 °C	900 °C	1000 °C	1100 °C	Attribution
Vibrational Bands (cm^−1^)	300.16					
	675.55					$\gamma -{Fe}_{2}O_{3}$
	1356.40					
		285.14				$\alpha -{Fe}_{2}O_{3}$
		401.93	290.15	290.15		
		603.81	605.47	605.47		
		1304.51				
	181.70	171.66				${Co}_{3}O_{4}$
	477.00	220.00	692.23	692.23		
	512.18	687.23				
					205.06	
			206.73	206.73	296.82	
		465.33	466.34	465.33	468.33	$Co{Fe}_{2}O_{4}$
			569.44	569.44	563.56	
					685.56	

**Table 3 molecules-28-07737-t003:** SAR results obtained for samples HT at 700 °C, 900 °C, and 1100 °C after a time of Δt=650 s.

Sample	ΔT(K);Δt=650 s	SAR (W/g)
700 °C	274.08 ±0.39	0.8 ±0.3
900 °C	276.32 ±0.77	2.1 ±0.4
1100 °C	278.78 ±0.24	3.6 ±0.2

## Data Availability

Data are contained within the article.

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
