# Peer review of "Cobalt Ferrite Synthesized Using a Biogenic Sol–Gel Method for Biomedical Applications"

_molecules, 2023, doi:10.3390/molecules28237737_

Round 1

Reviewer 1 Report

Comments and Suggestions for Authors

The article “Cobalt ferrite synthesized by a biogenic sol-gel method for biomedical applications" is dedicated to researching the physicochemical properties of cobalt ferrite. The obtained samples possess magnetic hyperthermia, presenting high saturation magnetization, biocompatibility and consisting only of cobalt ferrite. The research topic is relevant. The article can be published in the journal “Molecules” after making minor changes.

Line 80. The ratio of the initial components should be written. Why was powder of coconut water used for synthesis? How was this water obtained? I ask you to give its chemical composition.

Line 87. Please write the conditions for calcining the gel. The device that was used for thermolysis, please write.

The year of manufacture, the company, the country must be written for each device used. What material was the crucible made of for DTA? Which sample attachment was used? What is the gas flow rate?

The estimation of the error in determining cytotoxicity was carried out using which method? Cobalt ferrite was added in which part of the experiment to determine cytotoxicity? Unfortunately, this is not entirely clear. In what form was solid cobalt ferrite added? Probably some kind of solvent was used?

The temperature is indicated in Celsius in part of the article, and part in Kelvin. Please do it uniformly.

The direction of exo/endo effects is not indicated in Figure 1.

The small drawings (Figure 1, 6) are very fuzzy. They need to be rebuilt.

Line 161. Table 4.1 is indicated in the text of the article, and in the inscription under the table itself – “table 1". A similar situation is on line 282 – figure 4.2 is indicated in the text of the article, and in the inscription under the figure – “Figure 7". Please do it uniformly.

Band 1356.4 is indicated in Table 2. However, all other bands are indicated with precision to the second decimal place. Please do it uniformly.

Line 312. The sample that was obtained at 1100C showed better properties compared to which samples?

It is desirable to make the size of the conclusions smaller. Probably, some of the conclusions can be transferred to the discussion.

Thermogravimetric analysis (DTA/TGA) were conducted using the Hitachi STA7300. What equipment was used to develop the sample for physico-chemical analyses and cytotoxicity? After all, very little calcined sample remains in the crucible for thermogravimetric analysis.

Author Response

The authors appreciate all your suggestions and comments in order achieve a high level of quality of this manuscript. Please see the answers in the document attached.

Many thanks,

Sílvia Soreto Teixeira

Reviewer 2 Report

Comments and Suggestions for Authors

comments.

1.The abstract should be written by summarizing the problem, the method and the result

2The introduction needs to be improved with current advancement in the ferrite nanoparticles

3The motivation  and the novelty  is missing

4The introduction should be enriched with recent  ref ...1) scale up approach for the preparation of magnetic ferrite nanocubes and  other shapes with bench mark performance  for magnetic hyperthermia nature protocol2)A three fold increase in SAR performance for magnetic hyperthermia by compositional tuning in Zinc substituted iron oxide SP particles. local temperature  increment and induced cell death in intercellular magnetic hyperthermia.

3 The powder was heated from 500 to 11000 C, employing a heating rate 50C/min, there is no mention of  the duration  time, and why?. Was the furnace calibrated before the measurement.?

4Regarding the XRD data, the author has shown that the phase transition fro m Y-Fe2O3 to Co3 O4 to CoFe204 with transition temp frpm 5000C to 11000C. The discussion in this context is not appropriate and the given ref does not provide and strong evidence.

5The author should also do cationic distribution analysis and elemental analysis in the temp range 500 to 11000C.

6 The SEM images are  not clear to visualise the grain size. How did the author  have calculated the grain size? how many particles have been taken. how about the particles size probability  distribution curve. Also calculate crystallite size  using william hall plot to compliment.

7Regarding VSM measurement, calculate the  magnetocrystalline anisotropy which has strong  dependency on SAR result.. Pl follow the ref below

Tuning the magnetocrystalline anisotropy and spin dynamics in Co Zn 1-xFe2O4( 0<x<1) nanoferrite.

8 As the bare nanomagnetic particles tend to agglomerate in polar liquid such as water. both hysteresis effect and relaxation effect will  equally contribute to magnetic heating. The magnetic spin relaxation effect predominant in nano colloides.. Please explain heating mechanism, without ambiguity.

9 It  is recommended to perfrom in-vitro magnetic hyperthermia analysis with 2D and 3D cell model for better demonstration of the prepared nanoparticles.

10 summarize the major findings and future perspectives of the work in conclusion which missing.

Comments on the Quality of English Language

English to be improved 

Author Response

(The authors gave the same response as above.)
